# Internal Curing Effect and Compressive Strength Calculation of Recycled Clay Brick Aggregate Concrete

**DOI:** 10.3390/ma12111815

**Published:** 2019-06-04

**Authors:** Feng Chen, Kai Wu, Lijian Ren, Jianan Xu, Huiming Zheng

**Affiliations:** 1College of Civil and Transportation Engineering, Hohai University, Nanjing 210024, China; chenfeng160205@hhu.edu.cn (F.C.); 170204020002@hhu.edu.cn (L.R.); tonyhsu0226@hhu.edu.cn (J.X.); zhenghm19970419@hhu.edu.cn (H.Z.); 2Department of Civil and Environmental Engineering, National University of Singapore, Singapore 117576, Singapore

**Keywords:** recycled clay brick aggregate concrete, mixed recycled aggregate, replacement ratio, compressive strength, internal curing effect

## Abstract

In order to investigate the internal curing effect of recycled brick aggregate (RBA) in recycled aggregate concrete (RAC) and calculate its contribution to the final compressive strength, two RAC groups with different recycled aggregates and 6 replacement ratios (*r*) under 4 curing ages were tested. Results show that the compressive strengths of RACI and RACII decrease steadily with the increase of *r* when below 40%, and that there is a significant drop once the *r* is higher than 60%. The internal curing effect for RAC with a low RBA ratio is mainly reflected during the curing age of 14–21 days, while for RAC with a high RBA ratio, this internal curing effect appears earlier, during 7–14 days, and becomes very obvious after 14 days. In addition, the actual tested compressive strength of RAC replaced by 100% RBA exceeds around 40% of the expected compressive strength at the age of 28 days. When the age of RAC entirely with RBA is 28 days, the compressive strength caused by the internal curing effect accounts for around 28% of the actual tested compressive strength. The most appropriate *r* of RBA for RAC production is between 40% to 60%. Finally, the equations for calculating the compressive strength of RAC are presented considering the curing ages, the replacement ratios and the internal curing effect of RBA. Further, a unified equation is suggested for convenience in calculation.

## 1. Introduction

Using recycled materials to replace natural aggregate (NA) in concrete is important from the environmental aspect of reuse. Using crushed waste concrete and clay brick as alternative aggregates has particular significance since they can considerably reduce the problem of waste storage as well as help in the preservation of NA [1,2]. Therefore, recycled aggregate concrete (RAC) with waste concrete or clay brick as recycled aggregate (RA) is a green, interesting material with some unknown properties that need to be studied.

A few studies have contributed to the research into using crushed clay brick as the coarse aggregate in concrete production. Many of them have focused on the mechanical properties of RAC directly, especially its compressive strength [3,4]. The results of Ghernouti et al. [5] showed that it is possible to manufacture RAC blocks that are comparable in compressive strength to normal concrete, but with an appreciable reduction in weight. Miličević et al. [6] tried to find the optimal replacement ratio, and their experimental research illustrated that with a 50% fine and 45% coarse NA replacement ratio of crushed brick and tile, the compressive strength was only 18% lower than normal concrete. The investigation of Baradaran-Nasiri and Nematzadeh [7] again proved that there is no significant drop in the compressive strength of concrete with an up to 50% replacement ratio of RA. By recycling, it is possible to make concrete with higher strength compared with the in-situ strength of concrete of old structures [8].

Regarding various results from Guerra-Romero et al. [9] and Martínez-Lage et al. [10], it is suggested that the use of mixed RA is feasible only at the expense of minor losses of mechanical properties. However, their investigations only used two different RAs to replace NA, and mixed RAs are not used to completely replace the NA. Gonzalez-Corominas [11] set up five types of RAs: three coarse RAs sourced from parent concretes of 100, 60 and 40 MPa, as well as one coarse mixed RA and one fine ceramic RA.

Clay brick, as a kind of lightweight aggregate, can adsorb a certain amount of free water in the mixing process, which will turn into internal curing water and help with hydration in the later curing period. Thus, the hydration degree of the surrounding cement is significantly higher than the other cement paste, making concrete denser and reducing the possible generation of cracks [12,13]. The high water absorption of RBA is one of the most discussed parameters in terms of its application in the production of RAC, and it is considered as an alternative internal curing material. The experiment of Ondova and Sicakova [14] confirmed that at a constant amount of mixing water, along with the increasing water absorption of RBA, the final mechanical properties of the concrete are improved.

Currently, no systematic research achievements have been published regarding the internal curing effect of clay bricks in RAC, and there are no accurate equations for the calculation of compressive strength considering the internal curing effect. A few limited studies have been conducted on the viewpoint of RAC having low compressive strength at an early age but relatively high strength at a later age. Results obtained by some authors [15,16] indicated that the incorporation of lightweight aggregate decreases the early-age compressive strength up to 7 days, while the 28-day strength is close to the reference group. Durán-Herrera et al. [17] had observed a decrease of 12% in compressive strength for internally cured concrete up to 7 days, but no significant drop was found at later ages. Bentz [18] also found that internally cured mortars exhibited a higher compressive strength beyond 7 days. In the experimental investigations of Ge et al. [19], as the replacement level increased, the early age strength decreased, but as the curing age increased, the strength of concrete with clay-brick-powder was similar to the reference concrete, which shows its ability to enhance its later strength. Lei and Chen [20] also observed that fly ash had an adverse effect on the early compressive strength of RBA concrete but will slightly enhance later strength development.

Thus, to identify the internal curing effect of RBA and calculate the compressive strength of RBA concrete, this investigation characterizes two kinds of RAC cubes with six replacement ratios under four curing ages. Considering the internal curing effect of RBA, one initial equation and another unified equation for the compressive strength of RAC with RBA or mixed RA are proposed, respectively.

## 2. Materials and Methods

### 2.1. Materials

Portland cement (P.O. 42.5) with a *ρ*_c_ of 3100 kg/m^3^ is used as the binder. Normal river sand with a *ρ*_s_ of 2650 kg/m^3^ and fineness modulus of 2.75 is used as the fine aggregate for the mixtures. Coarse aggregates include NA, recycled concrete aggregate (RCA) and RBA. RCA and RBA are obtained from the abandoned concrete building and masonry walls near Hohai University, which are crushed and not pre-wetted for the mix process, as shown in Figure 1. The size of the uncrushed waste brick is 210 mm × 80 mm × 40 mm. The same series of sieves with sizes between 4.75–31.5 mm is used to ensure that the three coarse aggregates have the same continuous gradation. It should be noted that an aggregate fraction less than 5 mm is normally not used for RAC [21], so the sizes of most aggregates are between 10–30 mm. The grading curves of coarse aggregates are presented in Figure 2, which are in accordance with the maximum and minimum limits of Chinese standard GB/T 14685-2011 [22]. The properties (bulk density, apparent density, crushing index) of the coarse aggregates are tested according to Chinese standard JGJ 52-2006 [23] and the 24-h water absorption is tested separately according to GB/T 17431.2-2010 [24]. After soaking for 24 h, the water absorption of coarse aggregate is calculated according to Equation (1). Core drilling samples are taken from the parts with less stress or representative performance; then, the strength of random core drilling is tested according to Chinese standard JGJ T 384-2016 [25]. In summary, all properties are tested by three duplicate samples, and the average results are recorded in Table 1.
(1)$$w_{a}=\frac{m_{0}-m_{1}}{m_{1}}\times 100\%$$
where *w*_a_ is the 24-h water absorption of samples, *m*_0_ is the weight of the soaked sample, and *m*_1_ is the weight of the drying sample.

Compared with NA, the bulk density and apparent density of RA are lower, but the water absorption, porosity and crushing index are higher, which is in accordance with Laserna and Montero’s results [26]. The main reason for this is that there is a large amount of old cement mortar attached to the surface of RA. Also, the properties of RBA itself are much more different from NA than RCA due to its clay characteristic. By the random core drilling test, it is found that the strength of RBA is much less than the RCA.

### 2.2. Specimens Design

The specimens are designed to investigate the effects of the following parameters: (1) coarse aggregate type (or RAC type), including RACI (using NA as basic aggregate and RBA as RA) and RACII (using RCA as basic aggregate and RBA as RA); (2) the replacement ratio (*r*), meaning the volumetric percentage of recycled coarse aggregate to the total amount of coarse aggregate, including 0%, 20%, 40%, 60%, 80% and 100%; and (3) curing ages, including 7, 14, 21 and 28 days. Three duplicate cubic specimens (150mm × 150mm × 150mm) are prepared for each RAC group. The concrete mixes are designed according to JGJ 55-2011 [27], with a *w*/*c* ratio of 0.44 and sand ratio of 0.307, as shown in Table 2. In addition, the water absorption of the mixed coarse aggregates used in each specimen are also calculated according to Equation (1).

### 2.3. Experimental Mode

All concrete specimens were cast on the same day in a 50-L compulsory mixer. It was observed that with a higher replacement ratio of RA, the water absorption of the concrete mixtures is higher, and thus its workability loss during the pouring process was also higher. Although the water absorption of clay brick reduces the fluidity of concrete, it does not affect the concrete pouring in this test. In the mixing process, RBA absorbed part of the free water and stored it in pores, which was released to reach the unhydrated cementitious material in concrete and turn it into internal curing water to help with the continuous hydration in the later curing period. All specimens were watered each morning and evening to ensure a good external curing environment. Also, the internal curing of RBA was released to retain the internal relative humidity of concrete to ensure a sustained internal curing environment. The axial compression test was carried out with a 5000 kN digital hydraulic testing machine made in Changchun Testing Machine Factory (Changchun, China) according to Chinese standard GB/T 50081-2002 [28].

## 3. Test Results and Analysis

### 3.1. Curing Age

According to Chinese standard GB 50010-2010 [29], the compressive strength (*f*_ct_) of cylindrical specimens (Φ150 mm × 300 mm) converted from the average compressive strength of cubic specimens (150 mm × 150 mm × 150 mm) at ages of 7, 14, 21 and 28 days are used. The conversion results are shown in Figure 3, and with the same *r*, the *f*_ct_ of RAC gradually increased with the curing age, which is similar to the normal concrete.

For an *r* below 40%, the *f*_ct_ of RACI stays in the same strength grade with normal concrete at the same age, and also exceeds 30 MPa at 28 days. For an *r* between 60% and 80%, the *f*_ct_ of RACI is about 5 MPa lower than the normal concrete at the same age, so these RACs should be used after considering a strength reduction. For RACI-100, the *f*_ct_ decreases by 45.18%, 39.22%, 29.84% and 23.48% at 7, 14, 21 and 28 days when compared with normal concrete, respectively. 

RACII has a similar changing process in the *f*_ct_ with RACI due to the same major influential parameters of the RBA ratio and curing age. On the other hand, the RCA and NA are more similar in terms of material properties, and so the reasons for the failure of these two RACs are basically the same: (i) the RBA is crushed first due to the lower strength of random core drilling and higher crushing index than cement mortar, NA and RCA; (ii) the bond between RBA and cement mortar is poor, because the brick surface has a low adsorption capability for fresh cement paste, which can also be clearly observed during the mixing process; (iii) the fragile residual mortar on the brick surface will cause stress concentration under compression, accelerating the destruction of RAC; and (iv) the weak bonding area between the new mortar and the old one that attached to RBA will cause the separation of aggregates and cement mortar, which leads to shear failure in this area.

### 3.2. Replacement Ratio

A strength index (SI) is used as the relative compressive strength, which is defined as the ratio between the *f*_ct_ of RAC and the *f*_ct_ of normal concrete. According to Figure 4, the relation between SI and r is illustrated to reveal the effect of the RBA ratio on the compressive strength. The SI curves of RACI and RACII first drop slowly and then drop rapidly with the RBA.

The SIs of RACI-20, RACI-40, RACI-60, RACI-80 and RACI-100 decrease by 4.47%, 12.52%, 29.42%, 37.48% and 45.18% when compared with *SI* = 1 at the age of 7 days, which confirms that there is an obvious strength reduction at an early age of RAC with RBA. The main reason is that RBA as a kind of lightweight aggregate has a similar strength to the early hardened cement paste, so the concrete strength is only determined by the cement paste at an early age. Hence, there should be a certain amount of early-age monitoring and maintenance while using this RBA concrete. From RACI-20 to RACI-100, the decrease of SIs changes from 1.97% to 23.48% compared with *SI* = 1 at the age of 28 days; similarly, from RACII-0 to RACII-100, the decrease of the SIs changes from 4.01% to 23.48%. The compressive strength should be determined by the aggregate itself at 28 days since the strength of cement mortar turns out to be stronger than the aggregates. The SIs of RACI and RACII have a rapid growth at a later curing age as a result of the high porosity and strong water absorption of RBA. The absorbed water in the mixing process is released during the cement hydration, so the RAC can maintain a certain internal temperature, which helps the curing and strength growth. The SIs of RACI and RACII decrease steadily when the *r* is below 40%; then, there is a significant drop once the *r* is higher than 60%, when the RBA with a low strength becomes the main coarse aggregate in RAC.

### 3.3. Internal Curing Effect

The *f*_ct_ of RACI and RACII at each age period are displayed in Figure 5, and the percentage in the column means the ratio between the *f*_ct_ increment during a certain age period and the total 28-day *f*_ct_. As we know, the internal curing effect mainly results from the free water that was absorbed by light aggregate in the mixing process. This then proceeds to internal curing water and continues the cement hydration in the later curing period, which makes the concrete denser. The water absorption of RBA is much larger than that of RCA and NA, as shown in Table 2; thus, in order to study the internal curing effect of RBA, an assumption is made that the curing effect of RCA and NA is ignored. In general, free water exists adequately in concrete in the first 7 days, and so the internal curing effect from RBA is mainly reflected after 7 days [30]. First, in RACI-0, which has no RBA, the strength increment ratio of the four curing periods is 1: 1.86: 2.92: 3.69 compared with the *f*_ct_ from 0–7 days. Next, according to this ratio, the expected strength (*f*_ce_) for the other four groups can be calculated based on their 0–7-day *f*_ct_. This *f*_ce_, as shown in Figure 5 with the fold line, is the natural compressive strength without considering the internal curing effect of RBA. Also, the ratios of the *f*_ct_ and the *f*_ce_ in all curing ages are recorded in the chart below, which is a direct index for the internal curing effect of RBA.

As the *r* of RBA increases from 0% to 100%, the proportion of *f*_ct_ from the period of 0–7 days decreases from 27.1% to 19.4% for RACI, and decreases from 27.3% to 19.4% for RACII, which reflects the negative effects of RA on the strength of RAC. The proportion of *f*_ct_ in the period of 0–14 days decreases from 50.5% to 40.1% for RACI, and from 46.8% to 40.1% for RACII. However, the proportion of *f*_ct_ from the period of 14–21 days increases from 28.6% to 32.5% for RACI and from 27.4% to 32.5% for RACII, respectively. Similar results are obtained in the curing group of 21–28 days, which means that the proportion of *f*_ct_ gradually increases with the increase of RBA in the later curing period. The excess height of the column (*f*_ct_) to the line (*f*_ce_) can be considered as the contribution of the internal curing effect of RBA. When the *r* is less than 40%, the *f*_ct_ of RACI basically coincides with the *f*_ce_ at the age of 14 days, and for RACII, the *f*_ct_ is lower than the *f*_ce._ However, the *f*_ct_ / *f*_ce_ of RACI-40 and RACII-40 are up to 1.06 and 1.04 at the age of 21 days, indicating that the internal curing effect of RAC with a low RBA ratio is mainly reflected in the period of 14–21 days. When the *r* is more than 60%, the *f*_ct_ of RACI and RACII are higher than the *f*_ce_ at the age of 14 days by about 14% and 17%, while at the age of 21 days, the differences are much higher, at about 26% and 36% for RACI and RACII. At the age of 28 days, the differences are about 33% and 35%. It shows that the internal curing effect of the RAC with a high RBA ratio appears at the period of 7–14 days and appears obviously after 14 days.

The internal curing proportion represents the ratio between the strength gain from the internal curing effect and the total compressive strength. Thus, (*f*_ct_ − *f*_ce_) / *f*_ct_ represents the internal curing proportion at each curing age, as shown in Figure 6. *Δf*_ct_ or *Δf*_ce_ represents the compressive strength obtained within each curing period. So, (*Δf*_ct_ − *Δf*_ce_) / *Δf*_ct_ represents the internal curing proportion of the strength gain at each age period, as shown in Figure 7.

As shown in Figure 6, with the increase of age, the proportion of internal curing in the compressive strength increases gradually. When the age is 28 days, the proportion of strength brought about by the internal curing effect is 2.5%, 7.3%, 20.1%, 25.9% and 28.4% for RACI with an *r* of 20%, 40%, 60%, 80% and 100%, respectively. Similarly, the proportion is 1.25%, 19.9%, 28.8% and 28.9% for RACII with an *r* of 40%, 60%, 80% and 100%, respectively.

As shown in Figure 7, for RACI, the internal curing proportion gradually increases with the age period, while the internal curing proportion of strength gains reaches its peak at the period of 14–21 days for RACII, which shows that the cumulative internal curing proportion in Figure 6 at the period of 21–28 days basically remains the same. For RACI, when the *r* is above 60%, (*Δf*_ct_ − *Δf*_ce_)/*Δf*_ct_ are higher than 20%, and the maximum value of 45.5% is reached for RACI-100 at the period of 21–28 days. For RACII, when the *r* is above 60%, the (*Δf*_ct_ − *Δf*_ce_)/*Δf*_ct_ is usually more than 25%, and the maximum value ofa 42.94% is reached for RACII-80 at the period of 14–21 dys. As a result, the internal curing effect of RACII appears earlier and more obviously than that of RACI due to the use of two RAs in RACII.

As expected, a higher *r* of RA would introduce more strength reductions in RAC due to the brittleness of RBA, but the internal curing effect of RBA will compensate this reduction. The internal curing effect of RACI (RACII) begins to appear after using the RBA, and the *f*_ct_ / *f*_ce_ increases from 1.00 to 1.40 (1.00 to 1.41) with the change of *r* (Figure 5). The *f*_ct_ of RAC with high RBA ratio at 21 days is even equal to the *f*_ce_ at 28 days. Combined with the aggregate property and the internal curing effect, the *f*_ct_ decreases slowly when the *r* is below 40%, then has a sudden drop after the *r* is beyond 60%. So, despite the obvious internal curing effect after 60%, the strength loss of RAC is more serious. Therefore, in order to ensure the performance of RAC while using as much RA as possible, this paper considers a 40–60% replacement ratio of RBA as the optimal value for both RACI and RACII.

## 4. Calculation and Discussion

### 4.1. Calculation of Compressive Strength

To the best knowledge of the authors, no research effort has been devoted to investigating the internal curing effect of RBA in RAC, and there are no accurate calculations of compressive strength considering the internal curing effect. Thus, a new equation considering curing age, replacement ratio and internal curing effect is presented, which is appropriate for RAC with RBA. The specific equation is as follows:(2)$$f_{\mathrm{cc}}=\alpha \times \beta \times \gamma \times f_{\mathrm{ck}}$$
where *f*_cc_ is the calculated compressive strength of RAC (MPa), *f*_ck_ means the compressive strength of the ordinary concrete or RAC without RBA at 28 days (MPa), and *α* is the coefficient of curing age. Because the compressive strengths of RACI and RACII both increase linearly along with the curing age, it is reasonable to believe that curing age has a positive correlation with the compressive strength. Thus, *α* can be taken according to the strength increment at four ages (7, 14, 21 and 28 days) from RACI-0 and RACII-0 without considering the internal curing effect of RBA, as shown in Figure 8.
(3)$$\alpha =\left\{ \begin{array}{l} 0.0353\times d+0.0239\left(\mathrm{RACI}\right)\\ 0.0351\times d+0.0075\left(\mathrm{RACII}\right) \end{array}\right.$$
(4)$$\beta =\left\{ \begin{array}{l} -0.4487\times r+1\left(\mathrm{RACI}\right)\\ -0.4577\times r+1\left(\mathrm{RACII}\right) \end{array}\right.$$

*β* is the coefficient of the replacement ratio. Because the compressive strength decreases gradually with the replacement ratio, it is believed that they have a negative correlation. Thus, *β* can be taken according to the strength decrease under six replacement ratios (0%, 20%, 40%, 60%, 80% and 100%) at the age of 7 days without considering the internal curing effect of RBA, as shown in Figure 9.
(5)$$\gamma_{14d}=\left\{ \begin{array}{l} 0.1004\times r^{2}+0.0765\times r+1\left(\mathrm{RACI}\right)\\ 0.1104\times r^{2}+0.0792\times r+1\left(\mathrm{RACII}\right) \end{array}\right.$$
(6)$$\gamma_{21d}=\left\{ \begin{array}{l} 2\times (\gamma_{14d}-1)+1\left(\mathrm{RACI}\right)\\ 2.35\times (\gamma_{14d}-1)+1\left(\mathrm{RACII}\right) \end{array}\right.$$
(7)$$\gamma_{28d}=\left\{ \begin{array}{l} 2.5\times (\gamma_{14d}-1)+1\left(\mathrm{RACI}\right)\\ 2.35\times (\gamma_{14d}-1)+1\left(\mathrm{RACII}\right) \end{array}\right.$$

*γ* is the coefficient of aggregate type or internal curing effect. As mentioned above, the early compressive strength of RAC with RBA is low, but due to the powerful internal curing effect, the compressive strength increases at a faster rate after 14 days. Meanwhile, the internal curing effect is more obvious with a higher replacement ratio. Therefore, it is necessary to determine the *γ* according to the curing age and replacement ratio. Thus, *γ*_14d_ can be taken from the curve fitting results of the compressive strength ratio between 14 days and 7 days, as shown in Figure 10.

The comparison between the actual tested compressive strength *f*_ct_ and the calculated compressive strength *f*_cc_ can be seen in Figure 11. The ratio of *f*_cc_
*/ f*_ct_ is between 0.855~1.118, the maximum deviation is about 14.5%, and more than 89.58% of the data are within the deviation range of 10%. Furthermore, the *f*_cc_ / *f*_ct_ points to the ages of 21 days and 28 days basically being along the reference line of *f*_cc_ = *f*_ct_; thus, Equation (2) can reflect the compressive strength more accurately at the later curing age. In summary, there is a high degree of consistency between Equation (2) and the test results.

### 4.2. Unified Calculation Method

The two groups of calculation coefficients for RACI and RACII are inconvenient for practical application. According to Section 2.1, the properties of RCA are much more similar to NA than RBA, and the coefficients *α*, *β* and *γ* for RACI and RACII have a high degree of similarity to the results in Section 4.1. Thus, the unified coefficients *α*′, *β*′ and *γ*′ are used in Equation (8) as follows:(8)$$f_{\mathrm{cc}}{}^{\prime }=\alpha^{\prime }\times \beta^{\prime }\times \gamma^{\prime }\times f_{\mathrm{ck}}$$
where *f*_cc_′ is the unified calculated compressive strength of RAC (MPa), and *α*′ is the unified coefficient of curing age. Similarly, *α*′ can be taken from the fitting curve in Figure 12 based on the test results of RACI-0 and RACII-0.
(9)$$\alpha^{\prime }=0.0352\times d+0.0157$$
(10)$$\beta^{\prime }=0.4532\times r+1$$

*β*′ is the unified coefficient of replacement ratio. Similarly, *β*′ can be taken from the fitting curve in Figure 13 based on the test results of RACI and RACII at the age of 7 days.
(11)$$\gamma_{14d}{}^{\prime }=0.1054\times r^{2}+0.0779\times r+1$$
(12)$$\gamma_{21d}{}^{\prime }=\gamma_{28d}{}^{\prime }=2.35\times (\gamma_{14d}{}^{\prime }-1)+1$$

*γ*′ is the unified coefficient of aggregate type or internal curing effect. Similarly, *γ*_14d_′ can be taken from the fitting curve in Figure 14 based on the test results of RACI and RACII at the ages of 7 days and 14 days.

The accuracy loss from the unified calculation equation is extremely small (R^2^ is lower by less than 1%). As before, the comparison between the actual tested compressive strength *f*_ct_ and the unified calculated compressive strength *f*_cc_′ can be seen in Figure 15. The ratio of *f*_cc_′/*f*_ct_ is between 0.887~1.119, the maximum deviation is about 11.9%, and more than 93.75% of the data are within the deviation range of 10%. Furthermore, the *f*_cc_′/*f*_ct_ points for the ages of 21 days and 28 days are basically along the reference line of *f*_cc_′ = *f*_ct_; thus, Equation (8) can still reflect the compressive strength accurately at the later curing age. In summary, there is a higher degree of consistency between Equation (8) and the test results, and this can be used to calculate the compressive strength of RAC with waste brick as recycled aggregate in practical design.

## 5. Conclusions

The internal curing effect of RBA in RAC and the calculation of the compressive strength of RAC are analyzed in this investigation, and the following conclusions can be drawn:(1)The compressive strength of RAC with RBA or mixed RA gradually increases with the curing age but decreases with the *r* of RBA. The compressive strength of RACI and RACII decreases steadily when the *r* is below 40%; then, there is a significant drop once the *r* is higher than 60%.(2)The internal curing effect of RAC with a low RBA ratio is mainly reflected at the curing period of 14–21 days, and that of the RAC with a high RBA ratio appears at the curing period of 7–14 days, then becomes obvious after 14 days. The actual tested compressive strength of RAC replaced by 100% RBA exceeds around 40% of the expected compressive strength at the age of 28 days.(3)When the age of RACI-100 and RACII-100 is 28 days, the compressive strength caused by the internal curing effect accounts for 28.4% and 28.9% of the actual tested compressive strength. The internal curing proportion gradually increases with the age period for RACI, while it reaches its peak at the period of 14–21 days for RACII. (4)In order to ensure the performance of RAC while using as much RA as possible, a mixture with 40–60% RBA by volume is the most appropriate for RAC. From the viewpoint of compressive strength, it is feasible to completely replace NA with mixed RA.(5)The equations $f_{\mathrm{cc}}=\alpha \times \beta \times \gamma \times f_{\mathrm{ck}}$ of compressive strength for RACI and RACII are presented in this investigation according to the curing age and replacement ratio, also considering the internal curing effect of RBA. Further, the unified equation $f_{\mathrm{cc}}{}^{\prime }=\alpha^{\prime }\times \beta^{\prime }\times \gamma^{\prime }\times f_{\mathrm{ck}}$ presented is considered more suitable and can increase the convenience of the calculation. 

## 6. Notations


**Abbreviation**

**Full Name**

*d*
Curing age (d)
*f*
_cc_
Calculated compressive strength of RAC (MPa)*f*_cc_′Unified calculated compressive strength of RAC (MPa)
*f*
_ce_
Expected compressive strength without considering the internal curing effect (MPa)
*f*
_ck_
Compressive strength of ordinary concrete or RAC without RBA at 28 d (MPa)
*f*
_ct_
Compressive strengths of cylindrical specimens converted by the tested Compressive strengths of cubic specimens (MPa)NANatural aggregate
*r*
Replacement ratio of recycled coarse aggregate (%)RARecycled aggregateRACRecycled aggregate concreteRACIRAC with NA as basic aggregate and RBA as RARACIIRAC with RCA as basic aggregate and RBA as RARBARecycled brick aggregateRCARecycled concrete aggregate*w*/*c*Water / cement (-)
*SI*
A strength index (-)
*α*
Coefficient of curing age (-)*α*′Unified coefficient of curing age (-)
*β*
Coefficient of replacement ratio (-)*β*′Unified coefficient of replacement ratio (-)
*γ*
Coefficient of aggregate type or internal curing effect (-)
*ρ*
_c_
Apparent density of cement (kg/m^3^)
*ρ*
_s_
Apparent density of sand (kg/m^3^)Δ*f*_ce_Expected compressive strength within each age period (MPa)

## Figures and Tables

**Figure 1 materials-12-01815-f001:**
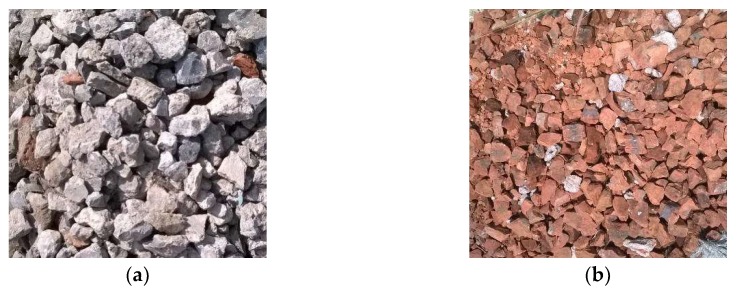
Recycled aggregates: (**a**) recycled concrete aggregate (RCA); (**b**) recycled brick aggregate (RBA).

**Figure 2 materials-12-01815-f002:**
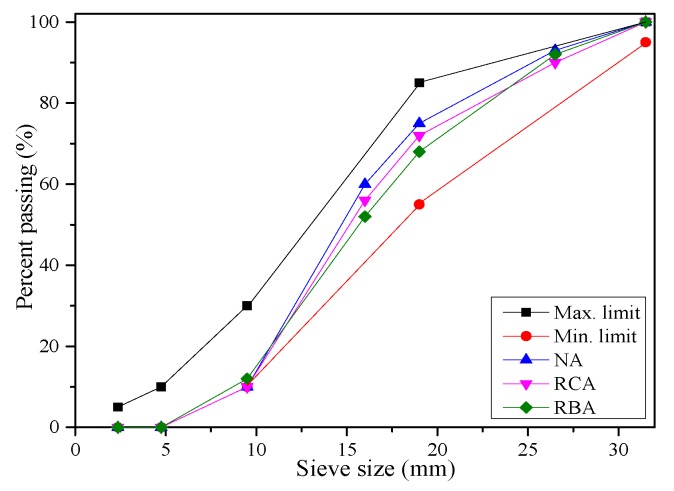
Grading curves of the natural aggregate (NA), RCA and RBA.

**Figure 3 materials-12-01815-f003:**
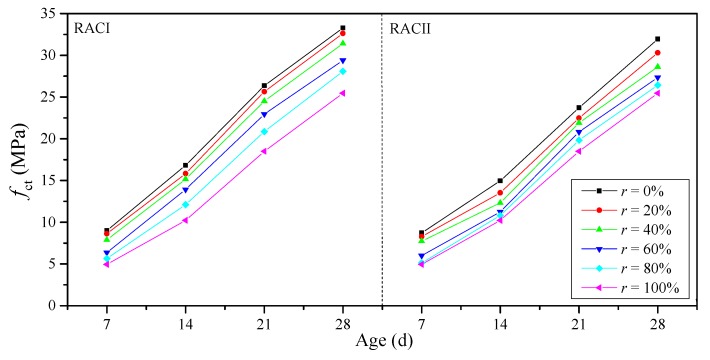
Compressive strength of RACs with age.

**Figure 4 materials-12-01815-f004:**
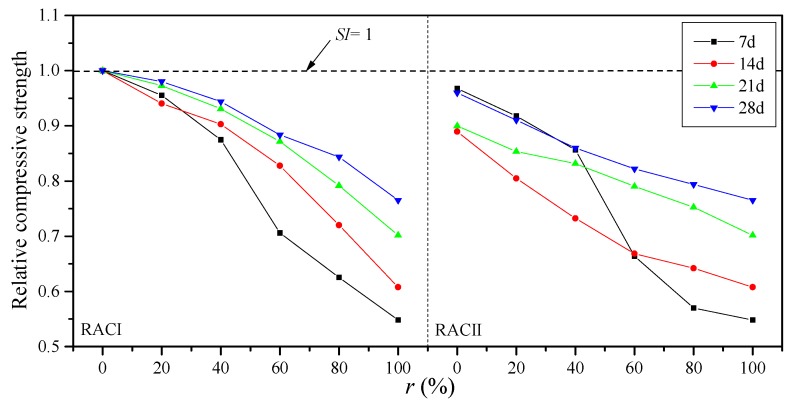
Relative compressive strength of RACs with different *r* values.

**Figure 5 materials-12-01815-f005:**
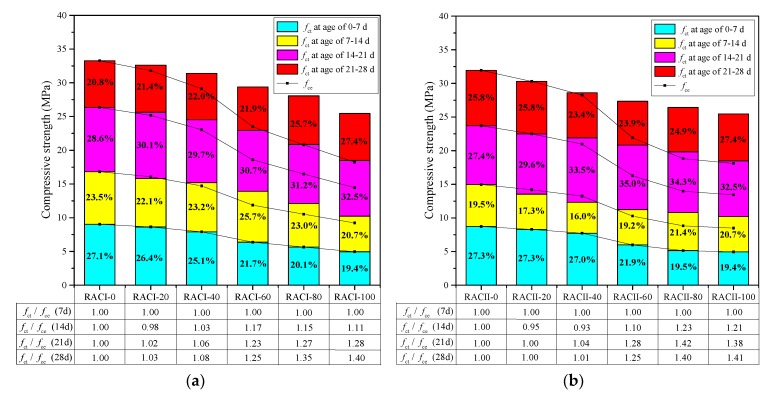
Compressive strength of RAC with RBA: (**a**) RACI; (**b**) RACII.

**Figure 6 materials-12-01815-f006:**
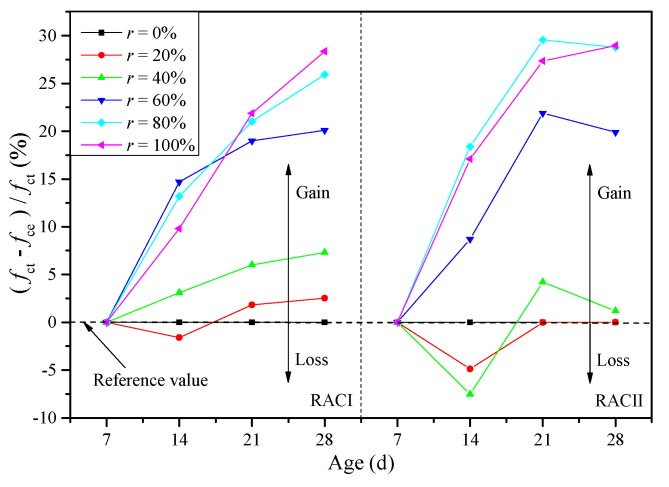
Internal curing proportion in each age.

**Figure 7 materials-12-01815-f007:**
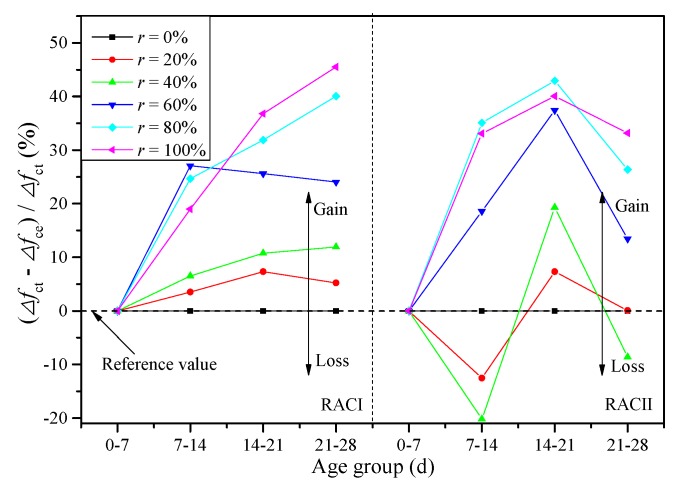
Internal curing proportion at each age period.

**Figure 8 materials-12-01815-f008:**
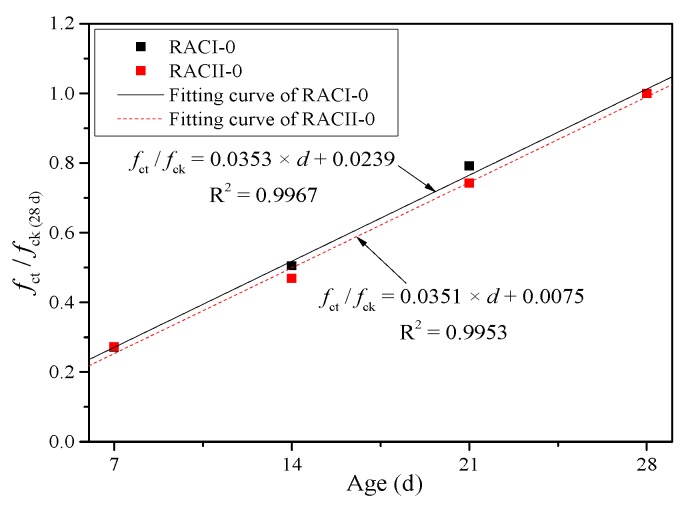
Fitting curve of coefficient *α.*

**Figure 9 materials-12-01815-f009:**
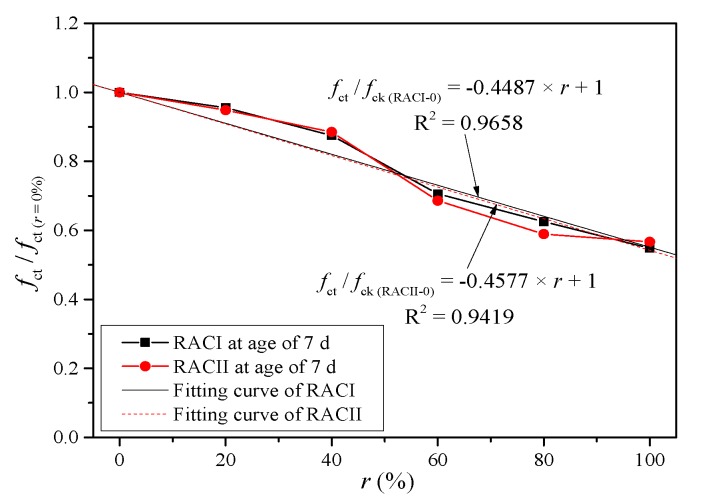
Fitting curve of coefficient *β.*

**Figure 10 materials-12-01815-f010:**
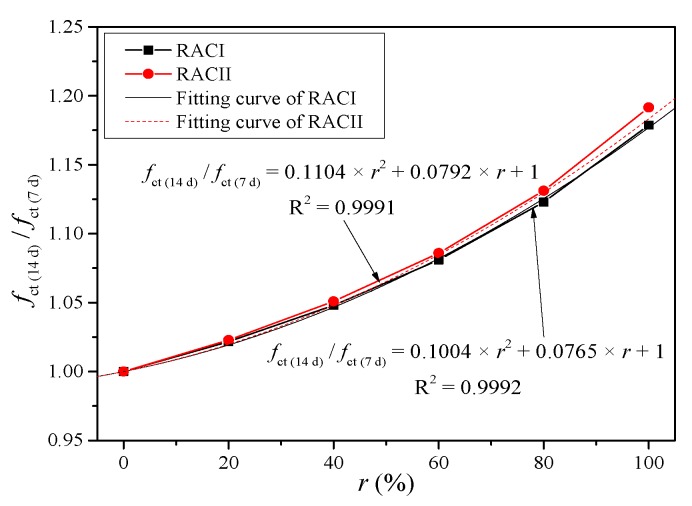
Fitting curve of coefficient *γ*_14d._

**Figure 11 materials-12-01815-f011:**
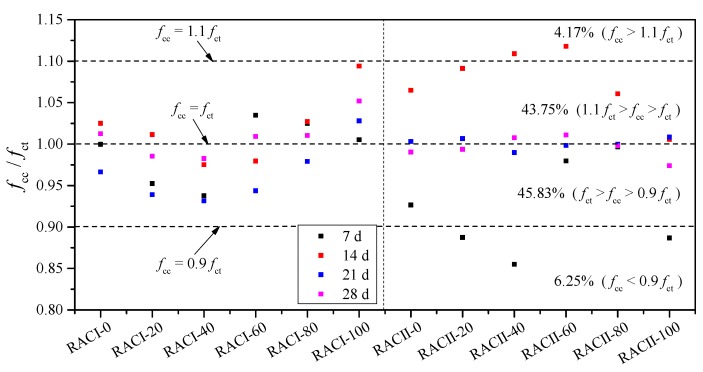
Comparison between the tested and calculated compressive strengths.

**Figure 12 materials-12-01815-f012:**
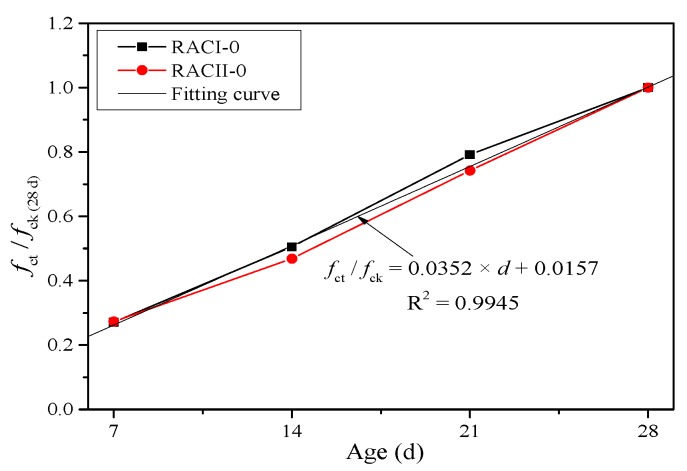
Unified fitting curve of coefficient *α.*

**Figure 13 materials-12-01815-f013:**
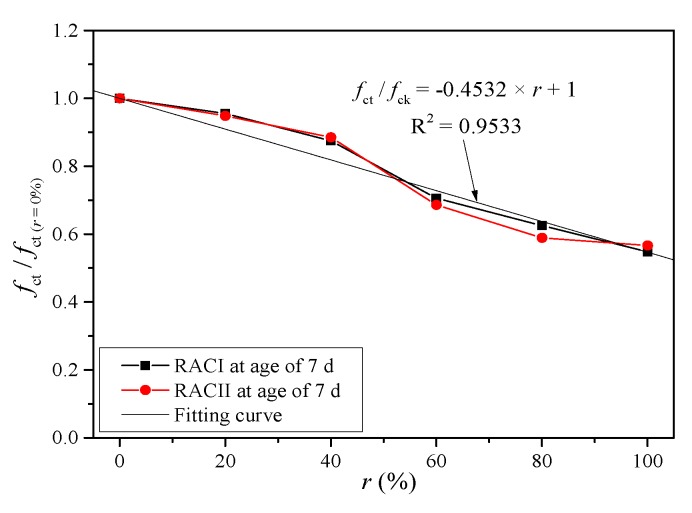
Unified fitting curve of coefficient *β.*

**Figure 14 materials-12-01815-f014:**
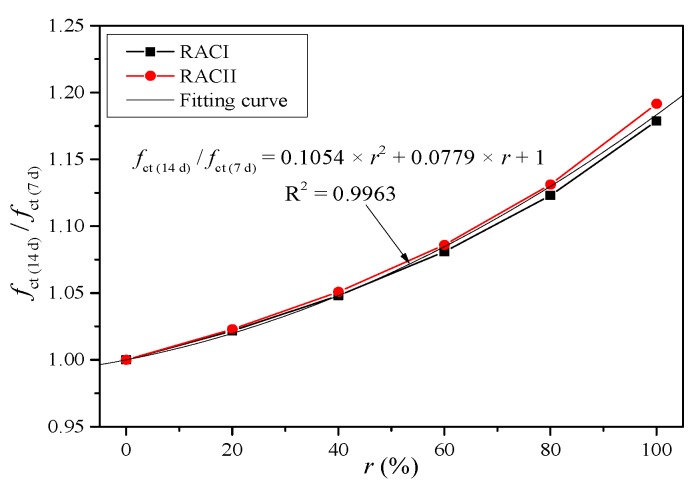
Unified fitting curve of coefficient *γ*_14d_.

**Figure 15 materials-12-01815-f015:**
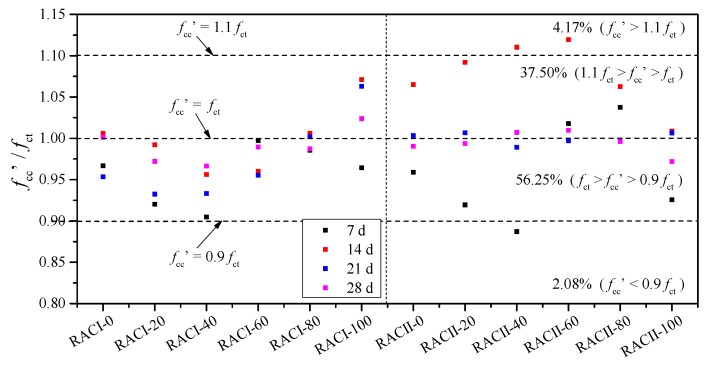
Unified comparison between the tested and calculated compressive strengths.

**Table 1 materials-12-01815-t001:** Material properties of coarse aggregates.

Aggregate Type	Continuous Grading (mm)	Bulk Density (kg/m^3^)	Apparent Density (kg/m^3^)	Water Absorption (%)	Crushing Index (%)	Random Core Drilling (MPa)
NA	10~30	1450	2800	0.3	1.5	
RCA	10~30	1389	2452	3.1	7.0	38~42
RBA	10~30	1280	1967	14.6	45.0	12

**Table 2 materials-12-01815-t002:** Mix proportion design of RAC specimens.

Specimen Group	Coarse Aggregates and Replacement Ratio	Water (kg/m^3^)	Cement (kg/m^3^)	Sand (kg/m)	NA (kg/m^3^)	RBA (kg/m^3^)	RCA (kg/m^3^)	Absorption (%)
RACI-0	RBA(0%), NA(100%)	185.00	420.45	574.01	1295.73	0.00	0.00	0.3
RACI-20	RBA(20%), NA(80%)	185.00	420.45	550.22	1056.63	185.57	0.00	3.0
RACI-40	RBA(40%), NA(60%)	185.00	420.45	525.64	808.10	378.46	0.00	5.7
RACI-60	RBA(60%), NA(40%)	185.00	420.45	500.00	549.57	579.11	0.00	8.5
RACI-80	RBA(80%), NA(20%)	185.00	420.45	473.31	280.43	788.00	0.00	11.3
RACI-100	RBA(100%), NA(0%)	185.00	420.45	445.50	0.00	1005.64	0.00	14.0
RACII-0	RBA(0%), RCA(100%)	185.00	420.45	523.41	0.00	0.00	1181.51	3.1
RACII-20	RBA(20%), RCA(80%)	185.00	420.45	508.55	0.00	191.77	956.21	5.3
RACII-40	RBA(40%), RCA(60%)	185.00	420.45	493.35	0.00	388.05	725.60	7.5
RACII-60	RBA(60%), RCA(40%)	185.00	420.45	477.78	0.00	589.01	489.49	9.6
RACII-80	RBA(80%), RCA(20%)	185.00	420.45	461.83	0.00	794.81	247.70	11.8
RACII-100	RBA(100%), RCA(0%)	185.00	420.45	445.50	0.00	1005.64	0.00	14.0

Note: The RACI-100 and RACII-100 mixes are the same.

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
