# Peer review of "Internal Curing Effect and Compressive Strength Calculation of Recycled Clay Brick Aggregate Concrete"

_materials, 2019, doi:10.3390/ma12111815_

Round 1

Reviewer 1 Report

The article is clear and well structured, but I have some doubts-comments:

·        It must be specified if the absorption of recycled aggregates has been considered. For example, has the water of the aggregate absorption been added to the water of hydration? Have they been pre-saturated before being incorporated into the mixer?

·        Only conclusions 4 and 5 can be considered as "conclusions". The rest is an analysis of obtained data. They could be improved if more general statements were stated as in the last two. Even so, in conclusion 5 the proposed equation should be stated.

Author Response

A word file (For Reviewer1—Authors' Response to Reviewers' Comments) is uploaded.

Reviewer 2 Report

In this study, internal curing effect and compressive strength of recycled clay brick aggregate concrete were investigated. In the literature, there is a lack of stduy on the internal curing effect of the recycled clay brick aggregate concrete. Therefore, this study has a potential of closure the gap on the above-mentioned subject.

This study needs some minor revisions to publish in this journal. The comments are as follows:

1) Is there any difference between internal curing and standard curing? Please mention more about the internal curing method in the "Materials and Method" section.

2) It should be noted that how the water absorption of the concrete were determined? Which method used?

3) The authors should be added to total weight of all concrete samples in Table 2. 

4) It should be stated the measured mix proportions rather than the theoretical mix proportions in Table 2.

5) How about the workability of the concrete mixtures? Because higher water absorption of the recycled clay brick aggregate, higher worability loss of the concrete mixtures. This should be mentioned in the manuscript.

6) In Figure 4 and 5, "compression strength" term should be replaced with "compressive strength".

Author Response

A word file (For Reviewer2—Authors' Response to Reviewers' Comments) is uploaded.

This manuscript is a resubmission of an earlier submission. The following is a list of the peer review reports and author responses from that submission.

Round 1

Reviewer 1 Report

-          On line 36 you do not indicate what the acronym RBA refers to (indicated in the abstract). However, the definition of the acronym RCA is repeated in the abstract and in the introduction. Put them one way or another, but always the same. I recommend doing it from the introduction.

-          Similarly, sometimes you use RA and other Ras (lines 53 or 89, for example). Homogenizes the format.

-          You don’t mention anything about Crushing index or Random core drilling. Indicate the standard that you have used and comment on the results obtained.

-          Line 112: Replaces "coase" with "coarse"

-          Although it can be deduced, indicate in parentheses what the acronyms RCAI and RCAII mean, the first time they are used (line 113)

-          Indicates if the absorption water of the recycled aggregates were considered in the manufacture of the concrete, and if so, in what way (saturating them previously, adding the absorption water ... etc)

-          The results of table 3 come from the breaking of test pieces of 15x15x15? If so, a conversion to Æ15x30 cylindrical specimens should be made

-          The data in Table 3 would be more illustrative if they appeared on a graph.

-          In section 4, it should be indicated if there is similar bibliography or not, comparing them where appropriate.

Author Response

Authors' response to Reviewer 1 can be obtained from the attachment.

Reviewer 2 Report

The paper presents original results of systematic investigation on substitution of natural aggregate by recycled brick aggregates. Although, there is no novelty, the paper is of interest and will enlarge a knowledge database on application recycled aggregates in concrete technology. However, the paper needs changes to make it clear and understable for readers:

1) The paper text should carefully verified from language point of view; there are various types of imperfection, e.g.  :

-   l. 65 „As far as I am concerned..”- the paper is submitted by three authors;

- incomplete sentences, e.g.: “However, a few researches on the effect of using crushed clay as a kind of coarse aggregate in concrete production.” or undefined parameters “Random core drilling”

2) Authors used to many presentation ways of results, i.e. table, figure and detailed description in the text, it makes the text in very complicated and  it is difficult to follow Authors’ way of thinking; Authors should decide which way is better but avoid repetition;

3) sophisticated analysis of substitution effect with brick aggregates is based on the changes in compressive strength; there are no information on: numbers of samples per concrete mix,  dispersion of strength results; what kind of compressive strength was used for analysis – average?.

This is important for reliability of analysis performed, e.g. detailed analysis of changes of compressive strength, especially in the relation to internal curing including data presented in the table 4.

4) internal curing effect: it could be worth to discuss this issue also quantitative way, there is only general discussion in the text.

5) Figures:

- Fig. 3: horizontal axes caption: better r – substitution rate instead symbols of sample groups.

- Fig.4 expected value is not defined, why it is called “expected value” in this chapter

- Fig 5: horizontal axes caption: this is not “cumulative age”

6) Chapter 4: calculation too complicated to have practical meaning

Author Response

Authors' response to Reviewer 2 can be obtained from the attachment.
